# Chemoenzymatic indican for light-driven denim dyeing

Gonzalo Nahuel Bidart [1], David Teze [1], Charlotte Uldahl Jansen [2], Eleonora Pasutto [1], Natalia Putkaradze [1], Anna-Mamusu Sesay[3], Folmer Fredslund [1], Leila Lo Leggio [4], Olafur Ögmundarson[5], Sumesh Sukumara [1], Katrine Qvortrup [2] ✉ & Ditte Hededam Welner [1] ✉

Blue denim, a billion-dollar industry, is currently dyed with indigo in an unsustainable process requiring harsh reducing and alkaline chemicals. Forming indigo directly in the yarn through indican (indoxyl-β-glucoside) is a promising alternative route with mild conditions. Indican eliminates the requirement for reducing agent while still ending as indigo, the only known molecule yielding the unique hue of blue denim. However, a bulk source of indican is missing. Here, we employ enzyme and process engineering guided by techno-economic analyses to develop an economically viable drop-in indican synthesis technology. Rational engineering of *Pt*UGT1, a glycosyltransferase from the indigo plant, alleviated the severe substrate inactivation observed with the wildtype enzyme at the titers needed for bulk production. We further describe a mild, light-driven dyeing process. Finally, we conduct techno-economic, social sustainability, and comparative life-cycle assessments. These indicate that the presented technologies have the potential to significantly reduce environmental impacts from blue denim dyeing with only a modest cost increase.

Indigo is the only known molecule yielding the unique and beloved hue of blue denim. It is a vat dye requiring harsh reducing agents such as dithionite and alkaline conditions for dyeing[1,2]. Indigo dyeing has significant negative environmental and social implications, including pollution of waterways and soil with concomitant crop loss, as well as human toxicity[3–6]. Indican, the β-*O*-glucoside of the indigo precursor indoxyl, is the natural storage form of indigo in indigo-producing plants[7]. It has been suggested as an environmentally mild blue denim dyeing agent since indican dyeing would eliminate reducing agents while still ending as indigo in the yarn[7–11]. To efficiently leverage indican in denim dyeing, two processes must be developed that have environmental benefits over the conventional method while still being affordable: a production process (Fig. 1A), and a dyeing process (Fig. 1B).

Here, we address these two aspects, considering the three dimensions of sustainability, i.e. (1) economic viability, (2) environmental performance, and 3) social impacts. To achieve economically feasible indican production, we engineer a stabilized glycosyltransferase mutant to convert indoxyl to indican at high substrate titers, a process with little environmental impact and no significant social impact. Further, we demonstrate economically feasible and low-impact enzymatic and light-driven denim dyeing processes, which have the potential to alleviate the current pressure on textile workers' health and livelihood.

## Results

### Chemoenzymatic synthesis of indican

To compete with the current blue denim dye, *i.e.* chemically synthesized indigo sold at 5 USD/kg, bulk indican production must be equally

[1]Novo Nordisk Center for Biosustainability, Technical University of Denmark, Kemitorvet 220, DK-2800 Kgs. Lyngby, Denmark. [2]Department of Chemistry, Technical University of Denmark, Kemitorvet 206, DK-2800 Kgs. Lyngby, Denmark. [3]Lab for Sustainability and Design, Designskolen Kolding, Ågade 10, DK-6000 Kolding, Denmark. [4]Department of Chemistry, University of Copenhagen, Universitetsparken 5, DK-2100 Copenhagen, Denmark. [5]Faculty of Food Science and Nutrition, University of Iceland, Aragata 14, 102 Reykjavík, Iceland. ✉e-mail: kaqvo@kemi.dtu.dk; diwel@biosustain.dtu.dk

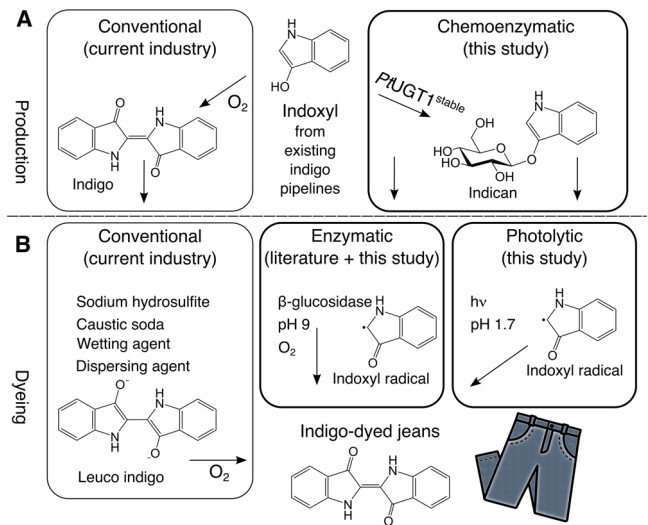

**Fig. 1 | Overview of indigo and indican production and use. A** production of indigo and indican in industry, literature, and this study. **B** dyeing processes with indigo (currently used in industry) or indican (proposed in literature and this study) resulting in both cases with blue denim represented by a blue jean.

affordable without introducing new strains on society or the environment. We estimate a raw material cost target of <12 USD/kg (see Methods). Given that cell-free biocatalysis, *i.e.* the use of enzymes in vitro, can be more scalable and efficient than cell factories[12], we investigated the three dimensions of sustainability associated with enzymatic synthesis of indican from indoxyl, captured from existing indigo production pipelines, by the natural indoxyl glycosyltransferase *Pt*UGT1[7]. *Pt*UGT1 is a soluble UDP-dependent glycosyltransferase (UGT)[13] found in the indigo-producing plant *Polygonum tinctorium*. Its limited stability in operating conditions was previously partly alleviated by immobilization but with severe activity loss[8]. Instead, we used a combination of consensus mutagenesis, loop grafting, and disulfide bridge engineering, to rationally introduce stabilizing mutations outside the active site of *Pt*UGT1 (Fig. 2, Supplementary Fig. 1, and Supplementary Table 1). Mutations with a favorable effect on melting temperature ($T_m$) were combined to arrive at several stabilized variants with retained wildtype activity. A decapuple combinatorial variant with best stability/activity combination was chosen for further work (*Pt*UGT1[stable]), displaying a $\Delta T_m$ of +13.1 °C (+13.8 kcal/mol[14]) (Fig. 2C, Supplementary Fig. 1C and Supplementary Table 1). The mutations' stabilizing effect can likely be attributed to increased rigidity due to increased P/G ratio (E75P, G222D, G430K), improved hydrophobic packing (S110V, I188L, T388A, GV296/297LG), as well as promotion of polar interactions (Q86K, S413K). *Pt*UGT1[stable] is robust in high substrate concentration, heat, organic solvents, prolonged storage (Fig. 3), and unbuffered water (Supplementary Fig. 1D).

We hypothesized that *Pt*UGT1[stable] can be leveraged to produce indican in bulk quantities from indoxyl in the existing industrial pipeline (Fig. 1). Indeed, *Pt*UGT1[stable] synthesized indican in vitro with a total turnover number (TON) > 250.000 in 24 h (Supplementary Fig. 1E). However, techno-economic assessment (TEA) initially indicated a prohibitively high raw material cost (470.220 USD/kg, Supplementary Fig. 2B). We projected that the process would be economically viable if 1) the co-substrate UDP-glucose could be recycled; 2) 60% conversion of at least 100 mM indoxyl could be achieved; and 3) the expensive HEPES buffer could be replaced with a pH stat (Supplementary Fig. 2B, theoretically optimized). For UDP-glucose recycling, we successfully implemented a sucrose synthase (SuSy)-based system[15–17]. We were able to demonstrate 65% conversion of 100 mM indoxyl in a cheaper phosphate-citrate buffer (Fig. 3A), resulting in a raw material cost of 15 USD/kg. Notably, due to a lack of

chemostability[18], *Pt*UGT1 wildtype is completely inactivated at this substrate concentration (Fig. 3A). On industrial scale, it will be useful if the reaction can happen in water with a pH stat instead of a buffer system, and we show here that *Pt*UGT1[stable] is functional in water (Supplementary Fig. 1D). We envisage the use of *Pt*UGT1[stable] and SuSy as lysates in solution, since previous attempts to immobilize these enzymes resulted in dramatic losses of activities[8,19]. With these data and projections, the raw material price reaches 9.9 USD/kg (Supplementary Fig. 2B, theoretical industrial process), or 5–13 USD/kg estimated by uncertainty analysis on the selling price of key raw materials (Supplementary Fig. 3G). The effect of process parameters such as yield, time, reactant amounts, and purchase prices, are illustrated by sensitivity and uncertainty analyses in Supplementary Fig. 3. From these analyses, even if the yield declines to 53% (Supplementary Fig. 3A) because of scale-up effects, or because of a desire to decrease reaction time to 20.1 h (Supplementary Fig. 3B), the process would be economically viable. We do not project a significant economic advantage in increasing enzyme concentration (Supplementary Fig. 3E). Overall, yield and reaction time are the cost drivers (Supplementary Fig. 3F), whereas realistic adjustments to UDP and sucrose concentrations (±50%) do not significantly affect the cost.

To avoid promoting a green-washing technology, we quantitatively assessed the environmental dimension of sustainability, employing a cradle-to-factory gate comparative life-cycle assessment (LCA) following the international reference Life Cycle Data system[20]. More than directly comparing to the existing industrial process, which is difficult when processes are at different technology-readiness levels (TRLs), we wanted to rigorously assess the projected environmental impacts. We found the majority (>90%) of end-point damages from indigo and indican production to stem from impacts on fossil resources, global warming, fine particulate matter formation, and terrestrial acidification (Fig. 4 and Supplementary Table 2). The major environmental gain from replacing indigo with indican is expected at the subsequent dyeing step, and we anticipated a slight increase in environmental impact of the production process due to the replacement of aeration with enzymatic treatment (Supplementary Fig. 2A). Indeed, we see a 1.2% increase, which can be monetarized[21] to 21.0 USD/kg versus 20.7 USD/kg (Supplementary Table 2) and mainly arises from ecosystem damages from marine eutrophication, water consumption, and land use (Fig. 4). This stems from enzyme manufacturing (of *Pt*UGT1[stable] and SuSy) and is commonly seen in biomanufacturing[22]. If the yield can be slightly increased from 65 to 66%, the environmental cost would be equal to that of conventional indigo (Supplementary Fig. 3C). Conversely, if reaction time and hence yield is decreased to the cut-off for economic feasibility identified above (20.1 h and 53%), the environmental impact will increase slightly, to 25.7 USD/kg. Supplementary Table 2 contains the full comparative LCA for all processes included in this manuscript, and LCA and TEA assumptions and sources can be found in Supplementary Table 3.

Finally, we considered social sustainability aspects of denim dyestuff production (A), an industry known to have adverse effects on human life quality[23], especially due to pollution of waterways. In fact, 17–20% of worldwide water pollution originates in the textile dyeing and finishing industry, according to the World Bank (Supplementary Note 1). Evidently, these aspects are interrelated with environmental issues. Quantitative social sustainability assessment is not a well-developed discipline, mainly due to the lack of clear definitions of indicators[24,25]. In addition, while textile dyeing is known to have severe toxicity problems[26], there is a lack of comparable quantitative data on the social impacts. Although regional case studies for dyeing hubs exist[27], these datasets are inaccurate for use on global scale. For these reasons we decided not to perform a quantitative life cycle analysis of the social impacts. Instead, we opted for a qualitative assessment of this technology's impact on social sustainability, *i.e.* the livability and health in production countries; We expect chemoenzymatic indican to

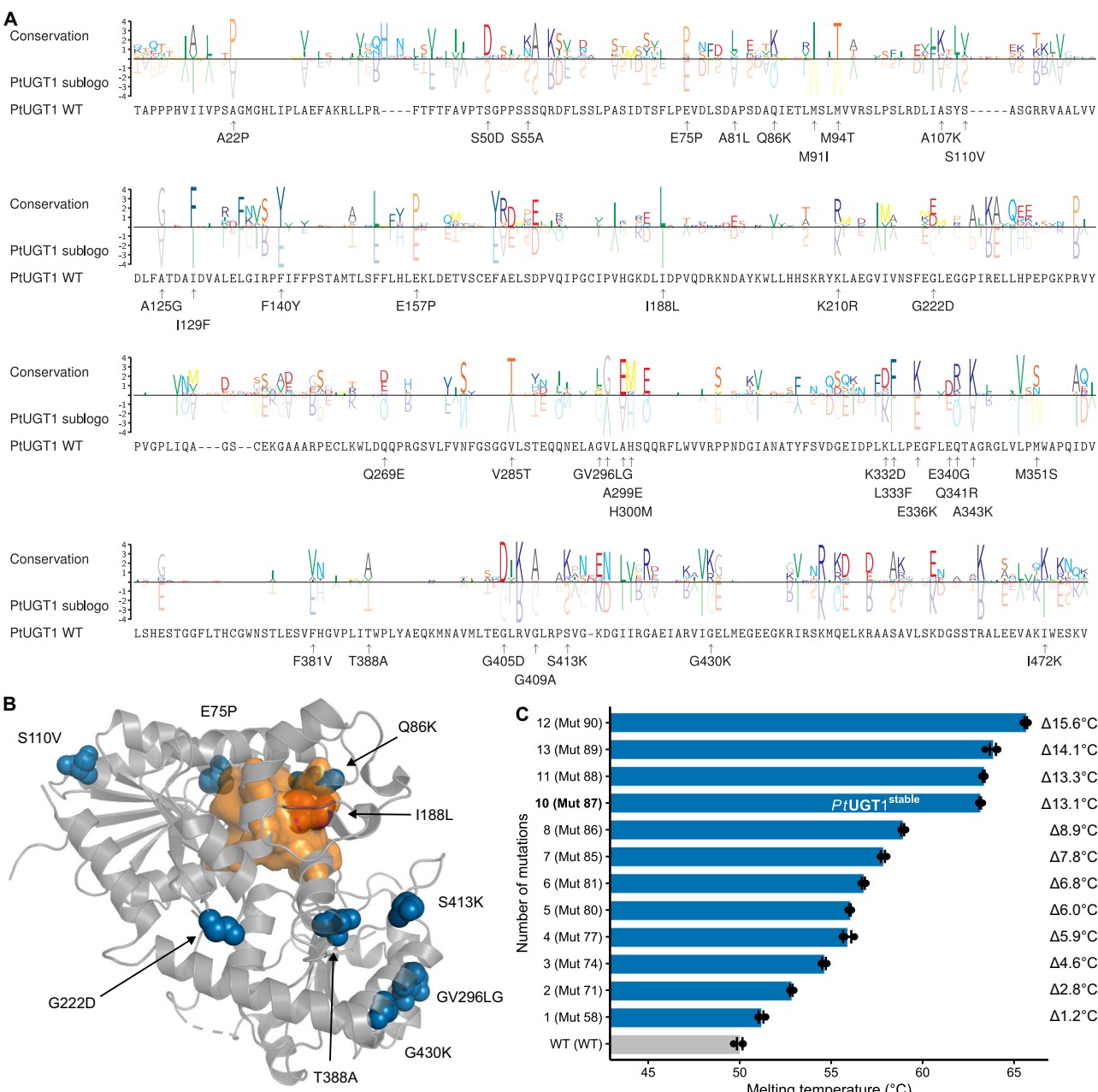

**Fig. 2 | Rational stabilization of PtUGT1. A** Multiple sequence alignment of PtUGT1 with 18 homologs, represented as differential sequence logos for PtUGT1 relative to conserved residues. Substitutions tested in vitro are annotated (see Supplementary Table 1). **B** Structure of PtUGT1 (PDB ID 5nlm) annotated with PtUGT1[stable] mutation sites (blue spheres). The active site is illustrated with an orange surface. **C** Melting temperature of PtUGT1 WT (grey bar) and variants (blue bars; source data are provided in Supplementary Table 1). Data are mean values ± SEM $n = 3$.

have a neutral effect on social sustainability aspects of dye production compared to conventional indigo production, since it will be a drop-in technology without harsh chemicals and associated health hazards. Further, the technology will likely not affect workplace location, and associated jobs along the supply chain will be retained. Our full assessment can be found in Supplementary Note 1.

## Dyeing with indican

Having established a chemoenzymatic route to indican, we investigated its use as a dye. Enzymes are already extensively used in the textile industry[28], and indican can be enzymatically cleaved to indoxyl, which can, in turn, be used as dyestuff by in situ generation of indigo[7,9–11]. The sustainability of this approach has not been quantitatively addressed. Several β-glucosidases catalyze the cleavage,

including *Secale cereale* *Sc*Glu[29] and *Thermotoga maritima* *Tm*Glu, as demonstrated in this study (Fig. 5A and Supplementary Fig. 4). We envisage a continuous process like the current vat-wetting and air oxidation process, but with the vat replaced by consecutive sprays of indican followed by β-glucosidase, as suggested by Hsu and colleagues[7]. A similar application method is industrially implemented for enzymatic textile bleaching (Lava Zyme NBF (Kaiser Tekstil)). With optimized reaction parameters (Supplementary Fig. 4), TEA estimates the enzymatic dyeing process to cost 0.2 USD per pair of jeans (Fig. 6 and Supplementary Fig. 2), *i.e.* 1.5-fold the cost of conventional dyeing, which is still low compared to the selling price of a pair of jeans. Sensitivity analysis indicates the cost to reach that of conventional dyeing by reduction of the dyeing volume from 3.2 to 1.9 liters, or by reduction of buffer from 20 to 11 mM, provided this will not reduce dyeing

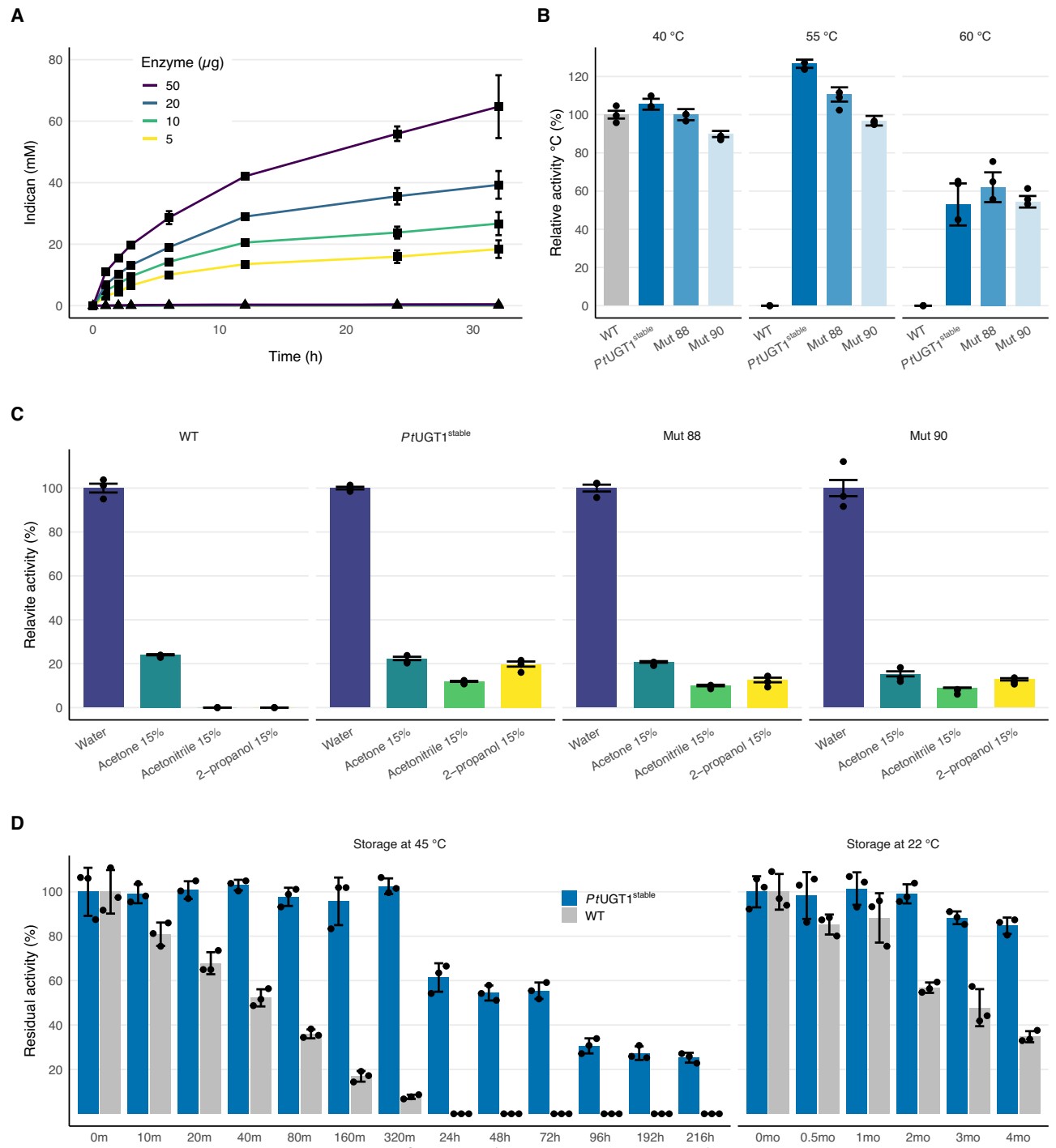

**Fig. 3 | PtUGT1^stable is chemically, thermally, and temporally stable. A** Indican synthesis at 100 mM indoxyl using 50 µg (purple), 20 µg (blue), 10 µg (green) or 5 µg (yellow) of PtUGT1 or PtUGT1^stable **B** Thermal tolerance was probed by measuring the relative activity of PtUGT1 (grey), PtUGT1^stable (dark blue), mutant 88 (blue) and mutant 90 (light blue) with increasing reaction temperature. **C** Solvent tolerance was probed by measuring the relative activity of PtUGT1 variants in acetone (blue bars), acetonitrile (green bars) and 2-propanol (yellow bars), to activity in water (purple bars),. Mut88 = PtUGT1^stable-F381V; Mut90 = PtUGT1^stable -F381V/A388C/A399C. **D** Residual activity of PtUGT1 (grey bars) and PtUGT1^stable (blue bars) after prolonged storage. Data are mean values ± SD n = 3. Source data are provided as a Source Data file.

efficiency (Supplementary Fig. 4E). β-glucosidase amount has minor cost impact. Considering the growing consumer demand for sustainable clothing[30], we deem this a commercially viable route. Importantly, we estimate an enzymatic indican denim dyeing process to have the potential to reduce the annual environmental cost of the denim dyeing process (B) by more than an order of magnitude (to 8%)

(Supplementary Table 2), even if a future industrial process would require 50% more enzyme or dyeing volume (Supplementary Fig. 4E). Remarkably, indican dyeing has the potential to reduce human lifetime loss, biodiversity loss, and financial loss relating to resource depletion to 9, 10, and 6%, respectively, of the losses associated with conventional dyeing (Fig. 6A). This is mainly due to global warming and fossil

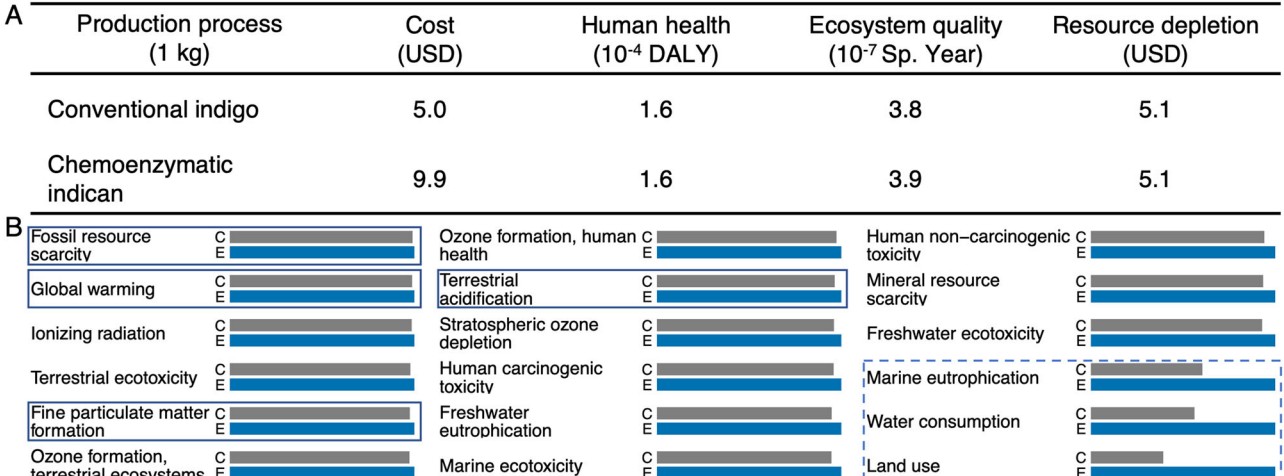

**Fig. 4 | Economic and environmental impacts of dyestuff production.**
**A** Endpoint damages[23]. DALY = Disability-Adjusted Life Years; Sp. Year = Species Year. **B** Detailed environmental impacts (mid points). C = conventional process for indigo production (grey bars), E = chemoenzymatic production of indican (blue bars). Impact categories contributing most to end point damages (>90%) are boxed. Impact categories contributing most to the increased impact of indican versus indigo are dash boxed. See Supplementary Table 3 for assumptions and data sources. Source data are provided in Supplementary Table 2.

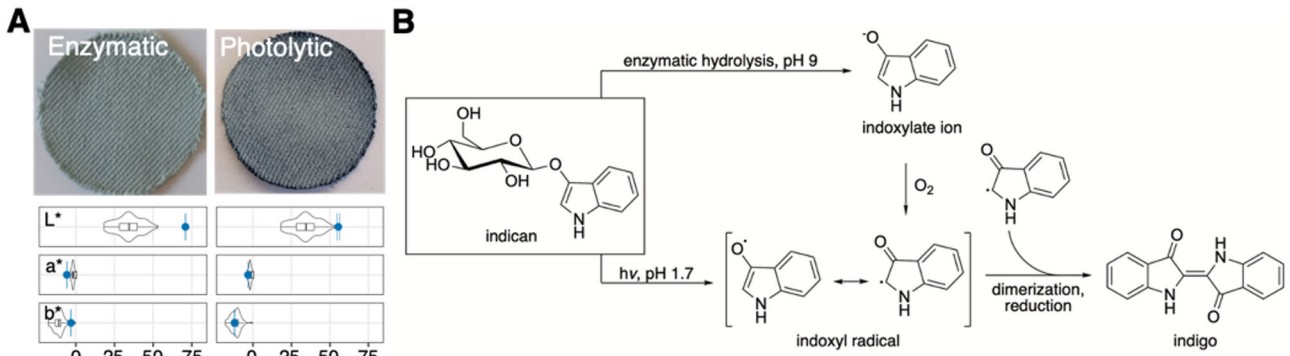

**Fig. 5 | Dyeing with indican. A** Denim dyed enzymatically (ScGlu) and photolytically (LED). Both methods produce blue denim, although photolytic dyeing seems to be more efficient since it leads to visually darker denim with the same indican amount. CIELAB color space (L* (dark-light), a* (green-red), and b* (blue-yellow)) analysis is shown below each swatch, with the values found in a selection of 60 samples of commercial blue denim (Supplementary Fig. 7) shown as violin and box plots for comparison. Data are mean values ± SD $n = 5$ (based on five measurements per swatch). Error bars indicate standard deviations. Center lines show the medians; box limits indicate the 25th and 75th percentiles as determined by R software; whiskers extend 1.5 times the interquartile range from the 25th and 75th percentiles, outliers are represented by dots. Source data are provided as a Source Data file. **B** Mechanisms of enzymatic and photolytic conversions of indican to indigo.

resource depletion being drastically reduced to 4 and 3%, respectively. On the other hand, marine eutrophication is more than doubled (228%) and water consumption is increased to 118%, due to enzyme manufacturing. This still accounts for only 0.03% of the total environmental impact of blue denim dyeing (Supplementary Table 2). Human toxicity and mineral resource scarcity are impacted by the manufacturing of the sodium phosphate used in the enzymatic dyeing buffer. It might be useful to consider an alternative to phosphate buffer for enzymatic indican dyeing in the future.

Given the environmental impacts associated with enzyme manufacturing[22], it is interesting to consider mild, non-enzymatic ways of converting indican to indigo. Since glucosyl halides and thioglycosides have been shown to undergo photolytic cleavage[31–33], we hypothesized that indican in combination with a light source instead of an enzyme can generate indoxyl radical, which readily would dimerize to indigo and dye denim (Fig. 5, Supplementary Figs. 5 and 6). Indeed, natural sunlight formed indigo in solution in a pH-dependent manner, and we could dye textile, also with a household light bulb (Supplementary Fig. 5A, B). While enzymatic indigo formation from indoxyl is

more efficient at alkaline pH and requires oxygen[11], the photolytic reaction has a pH optimum of 1.7 and seems unaffected by oxygen levels (Supplementary Fig. 5C-E) but dependent on light (Supplementary Fig. 5G). Below 20 mM HCl (pH 1.7), impurities are observed, likely due to acid hydrolysis of indican. This leads us to propose a mechanism (Fig. 5B), where photolysis of indican directly forms the indoxyl radical (Supplementary Fig. 5F). For comparison, the enzymatic process forms the indoxyl radical after deprotonation and oxidation of the enzymatically formed indoxyl[34]. Photolysis of indican follows first order kinetics, indicating suboptimal utilization of photons[35–37] (Supplementary Fig. 5G).

Using artificial light, we optimized the process (Supplementary Fig. 6A) to demonstrate efficient photolytic indican dyeing giving rise to a coloring comparable to commercial denim (Fig. 5A). However, photolytic dyeing requires longer time than enzymatic dyeing (see Supplementary Figs. 4C and 5G). Repeated dyeing dynamically increases color depth (Supplementary Fig. 6A), analogous to the repeated dipping in indigo baths currently employed in dyeing mills. However, the high electricity usage of the mixed wavelength artificial

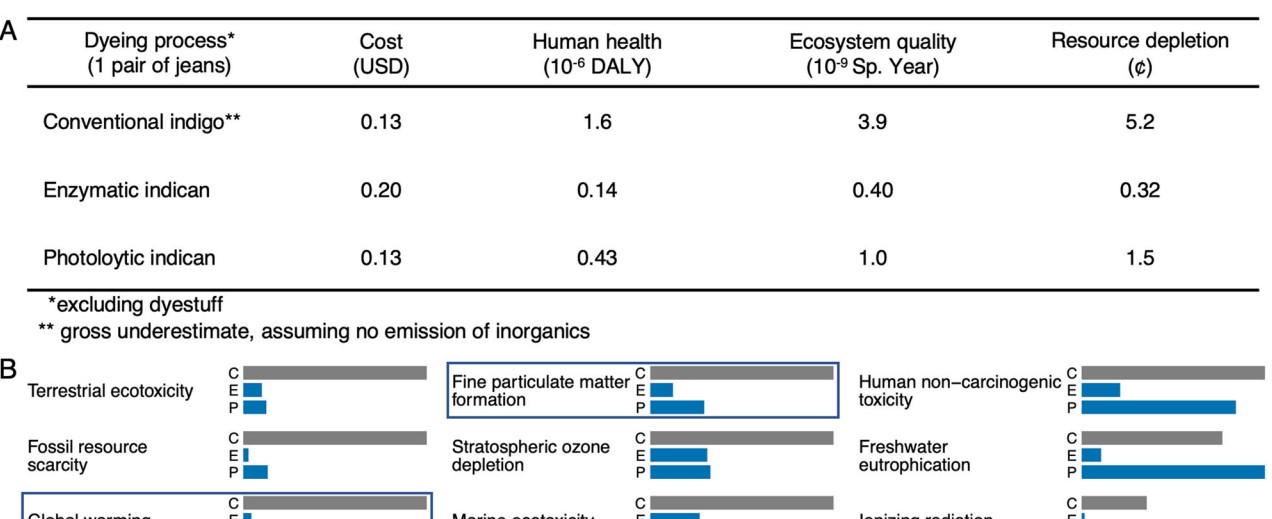

| Dyeing process* (1 pair of jeans) | Cost (USD) | Human health ($10^{-6}$ DALY) | Ecosystem quality ($10^{-9}$ Sp. Year) | Resource depletion (¢) |
|---|---|---|---|---|
| Conventional indigo** | 0.13 | 1.6 | 3.9 | 5.2 |
| Enzymatic indican | 0.20 | 0.14 | 0.40 | 0.32 |
| Photoloytic indican | 0.13 | 0.43 | 1.0 | 1.5 |

*excluding dyestuff
** gross underestimate, assuming no emission of inorganics

**Fig. 6 | Economic and environmental impacts of dyeing. A** Endpoint damages[23]. DALY = Disability-Adjusted Life Years; Sp. Year = Species Year. **B** Detailed environmental impacts (mid points). C = conventional indigo dyeing (grey bars), E = enzymatic indican dyeing (blue bars), P = photolytic indican dyeing (blue bars). Our LCA assumes complete wastewater treatment and worker safety in dyeing mills. Impact categories contributing most to end point damages (>90%) are boxed. See Supplementary Table 3 for assumptions and data sources. Source data are provided in Supplementary Table 2.

sunlight bulb is not economically or environmentally competitive (Supplementary Fig. 2 and Supplementary Table 2). Therefore, we investigated the dyeing capability of LEDs ranging from 308 to 660 nm in wavelength (Supplementary Fig. 6B). We observed 365 nm to give the highest conversion of indican to indigo. After optimizing the experimental setup (Supplementary Fig. 6C, D), we were able to achieve a coloring similar to the enzymatic process (Fig. 5A), for the same cost as conventional dyeing (Fig. 6A). We envisage the material to be dyed by soaking in an indican solution and then passing through an acidic aqueous solution in a shallow container covered by a LED panel of optimized wavelength, irradiance, and distance, for minimal electricity usage. The set-up could also be stacked similarly to vertical farms, to reduce the required area of the photodyeing mill. Accounting for potential scale-up effects, we estimate the price range to 0.08–0.14 USD per pair of jeans, depending on the required dyeing volume and electricity (varied +/− 50%, Supplementary Fig. 6F). With the current conditions, photolytic dyeing has the potential to reduce the environmental impact of blue denim dyeing with 73%, compared to a 92% reduction offered by enzymatic dyeing (Supplementary Table 2). Like for enzymatic dyeing, the reduction is mainly due to a potential for decreased global warming (to 17%, Fig. 6B). Although photolytic dyeing uses fewer mineral and land resources than enzymatic dyeing, the latter results in significantly less water and human toxicity, freshwater eutrophication, ionizing radiation, and water consumption. Optimizing the dyeing volume for minimal water consumption, as well as further reducing electricity usage, *e.g.* by using natural sunlight, might be relevant next steps. Photolytic dyeing would be environmentally competitive with enzymatic dyeing if the electricity usage can be reduced to 0.24 kWh per pair of jeans, *i.e.* to 60% of current experimental conditions (Supplementary Fig. 6F), while still being economically competitive with conventional dyeing. Interestingly, the economic cost is affected mostly by dyeing volume, whereas the environmental cost is affected mostly by electricity usage

(Supplementary Fig. 6F). It is worth mentioning that these impacts have been calculated based on the current European electric grid mix. Using 100% renewable energy would result in a significant decrease in environmental impact, a concept that is already applied to produce green hydrogen and ammonia.

Whether enzymatic or photolytic, we expect dyeing with indican to have a significant positive impact on livability and health in communities around dyeing mills, since the use of corrosive substances is eliminated, and the generation of toxic waste is significantly reduced, decreasing the pollution of groundwater, and sparing surrounding agricultural soil (Supplementary Note 1). Furthermore, the reduced environmental concerns can provide an incentive for more localized production in the Western denim market. This might enhance supply chain transparency, highlighted as one of the key factors to improving sustainability in the textile industry[38,39].

## Discussion

Using rational enzyme and process engineering in iteration with TEA, we have developed a biological drop-in technology based on $Pt$UGT1$^{stable}$ for economically viable production of indican in the existing indigo production pipelines (A). We then demonstrate that indican can be used to dye denim, either enzymatically or photolytically (B). We pioneer the application of sustainability assessments for processes at low TRL to direct biotechnological research, which we believe will prove to be key to fulfill the biotech promise. Considering the wider impact of the entire process (A + B) from dyestuff production to blue denim on a global scale, indican has the potential to be environmentally milder than and economically competitive with indigo, even without taking emissions of inorganics from conventional dyeing into account. Currently, 50.000 tonnes of indigo are produced yearly[7], primarily for denim dyeing. Replacing this large market with indican will require 320 tanks of 100 m³ run every second day year-round with the process parameters reported here. This will decrease

the yearly global $CO_2$ emission with 3.5 megatonnes, given 4 bn jeans traded yearly according to market analyses. Furthermore, workers in denim mills will no longer be exposed to harmful chemicals. This work demonstrates the power of integrating biotechnology and sustainability research to provide a direction for sustainable blue denim.

# Methods

## Materials

Laboratory chemicals were purchased from Sigma-Aldrich (USA), and CarboSynth (USA). Ready-to-dye denim was kindly provided by Nudie Jeans (Sweden). Specialized materials are detailed in the sections below.

## PtUGT1 variant design

For identification of mutation sites for consensus mutagenesis[40], a multiple sequence alignment was constructed for 18 enzymes with >60% sequence identity to PtUGT1 using Multiblast ClustalW2[41] (Fig. 2A). The positions where the original PtUGT1 residue was underrepresented were identified, and variants designed by the consensus approach (changing the residue to the most represented among homologs). Leveraging the structures of PtUGT1 (PDB ID 5nlm)[7], as well as homologs UGT72B1 (PDB ID 2vg8)[42], and UGT71G1 (PDB ID 2acv)[43], a rational analysis of the potential mutations was performed using Coot[44], mutations expected to have negative effect were omitted, and the final number of variants from consensus mutagenesis was 34. Identification of potential disulfide bridges was done with the software tools SSBOND[45] and Disulfide by Design 2.0[46]. The programs identified 35 residue pairs having the potential to form intramolecular disulfide bridges (Supplementary Fig. 1A), and 2 residue pairs having the potential to form intermolecular disulfide bridges. For loop grafting, the interdomain linker (residues 251–261 in PtUGT1) and loop C2[47] (residues 310–336 in PtUGT1) were analyzed in the structures of several homologs (PDB IDs 2vg8, 2acv, 6jtd[48], 6lf6[49], and 7q3s[50]). These two regions were chosen because of high B-factors in the PtUGT1 structure (PDB 5nlm[7]). Based on this, we grafted the interdomain linker from 6jtd, and loop C2 from 6lf6, both considerably shorter than the corresponding original PtUGT1 loops (Supplementary Fig. 1B).

## Cloning, expression, and purification of enzymes

Variants of PtUGT1 were constructed using the original expression vector (pTMH307) as template. The mutations were introduced by PCR with USER cloning (NEB, USA) using the primers synthesized by IDT (USA) listed in Supplementary Table 4. All constructs were verified by DNA sequencing service (Eurofins Scientific, Luxembourg) before transformation into chemically competent E. coli BL21 Star (DE3) (ThermoFisher Scientific, Germany) or SHuffle® T7 Express Competent (NEB, USA) in the case of disulfide bond-containing variants. Expression and purification of PtUGT1 WT and variants were performed as previously described[7] with minor modifications. 10 mL pre-culture was grown overnight in 2xYT media containing ampicillin (100 µg/mL) and used to inoculate 1 L cultures of 2xYT media with ampicillin (100 µg/mL). Cultures were grown at 37 °C in a MaxQ8000 incubator (Thermo Fisher Scientific, Germany) at 200 rpm and induced with 0.2 mM isopropyl-β-D-thiogalactopyranoside at OD600 ~ 1. Cultures were then grown at 18 °C for 21 h, and the cells were harvested by centrifugation. The cell pellets were resuspended in 50 mM HEPES pH 7.0, 300 mM NaCl, 40 mM imidazole pH 8.0. The cell suspension was lysed with 2 cycles through an Avestin Emulsiflex C5 homogenizer (ATA Scientific Pty Ltd., Australia) and treated with DNAse I (Merck, Germany). Cells debris was removed by centrifugation at 15.000 x g for 20 min at 4 °C. The cleared extracts were loaded onto 1 mL nickel affinity columns (HisTrap FF, Cytiva, USA) and the protein was purified using an Äkta FPLC system (Cytiva, USA). After washing the column with buffer (50 mM HEPES pH 7.0, 300 mM NaCl, 40 mM imidazole pH 8.0), elution was carried out with a 40–500 mM imidazole gradient. The peak fractions were analyzed with SDS-PAGE using NuPAGE 4–12% Bis-Tris Protein Gels (Thermo Fisher Scientific, USA) stained with Instant Blue (Expedeon Ltd., UK). Selected fractions were pooled, concentrated using a 50.000 MWCO Amicon Ultra-15 Centrifugal Filter Unit (Merck, Germany) and stored in 25 mM HEPES pH 7, 50 mM NaCl, 1 mM DTT. Final protein concentrations were determined by absorbance measurements at 280 nm using a ND-1000 spectrophotometer (ThermoFischer Scientific, Germany).

## Differential scanning fluorimetry (DSF)

Melting temperatures ($T_m$) of PtUGT1 WT and variants were measured by DSF using the Protein Thermal Shift Dye Kit (ThermoFisher Scientific, Germany) and a QuantStudio5 qPCR machine (ThermoFisher Scientific, Germany). Dye solution (1000x) was diluted to final (2x) in Buffer 2x (100 mM HEPES pH7, 100 mM NaCl). 10 µL of dye solution 2x was mixed with 10 µL of 0.8 mg/mL protein sample and pipetted in the qPCR multi-well plate. The plate was centrifuged 30 s at 142 x g and transferred to the qPCR machine. The protocol initiates with 2 min incubation at 25 °C, followed by a temperature increase of 0.05 °C/s up to 99 °C, and a final incubation of 2 min at 99 °C. Measurements were carried out in triplicate. Raw data was obtained with QuantStudio Design & Analysis Software 1.5.1 and further analyzed with Protein Thermal Shift™ Software v1.4 (Fig. 2C).

## Relative activity experiments

Measurement of PtUGT1 variant activities relative to WT activity was performed in triplicate with end point measurements of product formation using the model substrate 3,4-dichlorophenol (DCP). Product formation was monitored with reverse phase HPLC, using an Ultimate 3000 Series apparatus (ThermoFisher Scientific, Germany) and a Kinetex 2.6 µm C18 100 Å 100 × 4.6 mm analytical column (Phenomenex, USA). MilliQ water and acetonitrile containing 0.1% formic acid were used as mobile phases A and B, respectively. For the thermal stability experiment (Fig. 3B) PCR strip tubes containing 200 µl of reaction mixture (1 mM UDP-glucose, 50 mM citrate phosphate buffer pH 7, 500 µM DCP, 1 µg enzyme) were incubated at either 40 °C, 55 °C or 60 °C for 10 min. Reactions were stopped and analyzed at 290 nm using a multi-step program (starting at 5% B, ramp up to 25% B at 1.5 min, ramp up to 30% B at 3.5 min, ramp to up 100% B at 6.25 min, stay at 100% B until 7 min, gradient decrease to 0% B at 8 min, stay at 0% B until 9 min). Peak integration and data handling was performed in the Chromeleon software (ThermoFisher Scientific, USA). For the buffer minimization experiment (Supplementary Fig. 1D), the reaction mixture (1 mM UDP-glucose, 500 µM DCP, 1 µg enzyme) was incubated in either 44 mM, 5 mM, or 1 mM citrate phosphate buffer pH 7.0, or pure MilliQ water, at 40 °C. Samples were taken at 10, 20, 30, 60 and 120 min, and analyzed as described but using data from 240 nm. For the temporal stability experiment (Fig. 3D) freezer stocks of enzyme in buffer (100 mM HEPES pH7, 100 mM NaCl) were incubated either at 22 °C or 45 °C for different period of times, and their residual activities were determined as described and compared to their initial activity, i.e. before the incubation step. For the solvent tolerance experiment (Fig. 3C), the reaction mixtures where added the indicated solvents and relative activities were analyzed as described and compared to the activity in the absence of solvent.

## Indican production by PtUGT1 WT and PtUGT1<sup>stable</sup>

Since we did not have access to a stable and pure indoxyl preparation, the proof of concept for the synthesis of indican from high substrate concentration (Fig. 3A) was done with indoxyl acetate (100 mM) in combination with Esterase from Bacillus subtilis (Sigma Aldrich, USA). The reactions were performed in triplicate in an anaerobic chamber, using glass HPLC vials stirred with small magnets at 30 °C. Reaction mixtures consisted of 3.5 mg indoxyl acetate, 90 mM citrate

phosphate buffer pH 8.0, 1 mM UDP, 200 mM sucrose, 2U Esterase, and varying amounts of *Pt*UGT1 and *Glycine max* SuSy[51] at a molar ratio of 1:55, in 200 μL reaction volume (assuming that the delay in acetic acid release from indoxyl acetate is insignificant, due to the high esterase activity, the final pH is 6.55 for the majority of the reaction). The reactions were started by the addition of all three enzymes and the progression was followed by HPLC as described above. To estimate the total TON achievable with *Pt*UGT1[stable] in 24 h, indican formation was measured at *Pt*UGT1[stable] limiting conditions (Supplementary Fig. 1E). The reactions were performed in duplicate in an anaerobic chamber, using glass HPLC vials stirred with small magnets at 30 °C. Reaction mixtures consisted of 3.5 mg indoxyl acetate, 30 mM citrate phosphate buffer pH 7.0, 1 mM UDP-glucose, 100 mM sucrose, 2U Esterase, 695 μg SuSy, and varying amounts of *Pt*UGT1[stable], in 1.8 mL reaction volume. One extra reaction with half SuSy amount was set in parallel as control to confirm *Pt*UGT1[stable] limiting conditions. Reaction mixtures were incubated overnight prior to starting the reactions by the addition of UDP-Glc, *Pt*UGT1[stable] and SuSy. The progression was followed by HPLC as described above.

### Indigo formation from indican by ß-glucosidases

*Sc*Glu was expressed in *E. coli* and purified following the same procedure used for the *Pt*UGT1 variants described above. *Tm*Glu was purchased from Megazyme (Ireland). For the initial test (Supplementary Fig. 4C), enzymatic indigo formation in aqueous solution was visually assessed in Eppendorf tubes containing 40.5 μL of 150 ng/μL *Sc*Glu, 5 mM indican in water, 85 mM citrate phosphate buffer (pH values 7 and 8), 85 mM glycine/lysine (pH values 8, 9 and 10), or 0.1 M NaOH (pH value 13), respectively. For quantification of the deglucosylation (Supplementary Fig. 4A, B) reactions were performed in 100 μL final volume consisting of 75 mM citrate phosphate buffer pH 8.0, 1 mM indican, and varying amounts of a β-glucosidase, either *Sc*Glu (0.4–2 μg), or *Tm*Glu (0.7–7.9 μg, Megazyme). The reaction mixtures were incubated at 40 °C for 12 min and the reactions where then quenched either by heat (95 °C for 2 min, *Sc*Glu) or by addition of 0.5 M NaOH (*Tm*Glu). The samples were centrifuged in a tabletop centrifuge (Mini Star, VWR, USA) for 10 min at 11000 x *g* and the soluble fraction was analyzed by HPLC as described above (see relative activity experiments). Indican was quantified using the corresponding peak area and an indican calibration curve. Enzymatic indigo formation on textile (Fig. 6A and Supplementary Fig. 4D) was assessed on circular swatches of ready-to-dye denim of 20 cm² (5.04 cm diameter, 798 ± 2 mg) in a petri dish. These were dyed with 0.05 mg/cm² *Sc*Glu and varying amounts of indican (1–5 μmol/cm²) in 50 μL/cm² of 50 mM citrate phosphate buffer pH 9 (final concentration 16.7 mM) for 4 h at room temperature without stirring. After dyeing, the swatches were washed with water and detergent and dried before CIEL analysis (see below).

### Solution studies of photolytic indican cleavage

For the initial test (Supplementary Fig. 5A), indican (Sigma Aldrich, USA) dissolved in MilliQ water to different concentrations were incubated in the windowsill (exposed to sunlight through window glass) with or without foil wrapping for 24 h (approximately 4 h of direct sunlight, and 12 h of daylight). The samples were then visually analyzed for indigo formation (blue). For the solvent study (Supplementary Fig. 5C), 5 mg of indican (Sigma-Aldrich, USA) was dissolved in 520 μL either 80 mM HCl, 80 mM NaOH, milli-Q water, 100% dimethyl sulfoxide (DMSO), 100% methanol (MeOH), or 100% acetonitrile (ACN) in a glass container which was then closed with a glass lid and placed under artificial sunlight, using a 300 W Ultra-Vitalux bulb (Osram, Germany) for 2 h. The same setup was used for the quantitative studies (Supplementary Fig. 5D, E, G), which were conducted in 20 mM HCl unless otherwise noted. UPLC/MS analyses were done on an AQUITY UPLC system (Water, USA), equipped with PDA and either a SQD or a SQD2 electrospray MS detector. The column was an Accucore C18 2.6 μm,

2.1 × 50 mm (ThermoFisher Scientific, Germany). The column temperature was 50 °C, and the flow rate was 0.6 mL/min. The mobile phases A and B were 0.1% formic acid in water and in acetonitrile, respectively. The method involved a gradient from 5% B to 100% B in 3 min, then hold for 0.1 min, with a total run time of 5 min. For the oxygen dependency experiment (Supplementary Fig. 5D) an oxygen-deprived solution was prepared by bubbling 20 mM HCl (aq) with dry nitrogen gas for 85 min before the addition of indican. The container was then closed using a glass lid with high-vacuum grease to seal, illuminated for 2 h, and analyzed by LC-MS. For the pH dependency experiment (Supplementary Fig. 5E), the solvents were 80, 40, 20 10, or 0 mM HCl in milli-Q water, and the samples were illuminated for 6 h before LC-MS analysis. For the radical scavenger experiment (Supplementary Fig. 5F), gallic acid (5 eq., 85 μmol) was added to 20 mM HCl (aq), and the sample was illuminated, alongside a control without gallic acid, for 2 h, followed by visual inspection. For determining kinetics (Supplementary Fig. 5G), the reaction was done in the presence of ready-to-dye textile similarly as described below. A 70 μL sample was taken from the solution at different time points and analyzed by LC-MS. Decay of indican was followed by the area under the indican peak in the UV-spectrum of the sample compared to the peak area at time point 0.

### Photolytic dying with indican

For initial testing of sunlight (Supplementary Fig. 5A), a circular swatch of ready-to-dye denim (diameter 3.7 cm) was dyed in the windowsill (3 h of constant sunlight + 3 h of cloudy weather) using 1.6 mL of 40 mM indican in MilliQ water. The swatch was kept moist by the regular addition of water. It was then washed, dried, and photographed. Preliminary experiments with a household lightbulb (Supplementary Fig. 5B) were carried out in PCR tubes containing squares of untreated stout cotton (140 g/m², Stof2000, Denmark) of approx. 0.25 cm × 0.5 cm and 100 μL of MilliQ water, 100 mM lysine pH 9.0, or 100 mM glycine-NaOH pH 11, and varying amounts of indican or water (negative control). The tubes were placed under a household light bulb (LEDARE LED, 75 W, 1055 lm, IKEA (Denmark)) over night. The distance from samples to the bulb was approximately 8.5 cm. After dyeing, the squares were washed with water for 5–10 min at 95 °C and photographed next to the remaining liquids in the tubes. A circular swatch of the stout cotton (diameter 3.7 cm) was likewise dyed with the household bulb, using 1.6 mL of 40 mM indican in water for 2.5 days in a petri dish. The shortest distance from the sample to the bulb was approximately 8.5 cm. After dyeing, the swatch was washed with hot water (95 °C) and dried at 65 °C prior to photography. For photolytic dyeing with artificial sunlight (Fig. 6A and Supplementary Fig. 6A), a 300 W Ultra-Vitalux bulb (OSRAM) was used. One μmol/cm² indican was dissolved in 20 mM HCl (aq) in a glass container slightly larger than the fabric. The fabric was completely covered by the solution. The container was covered with a glass-lid and placed under the light source. After dyeing, the fabric was rinsed with water, air-dried, and photographed. For the repeated dipping experiment (Supplementary Fig. 6A), the volume of the HCl solution was 0.23 mL/cm² and the textile was dyed for 6 h with a lamp distance of 30 cm, while stirring with a magnet. For probing mixing methods (Supplementary Fig. 6A), two dyeing cycles of dipping in indican solution followed by 6 h exposure each was employed. The lamp distance was 30 cm, and there was either no mixing, mixing on a laboratory magnet stirrer, or on a laboratory platform shaker providing environmental shaking. The volume of the HCl solution varied from 0.23 ml/cm² to 0.45 mL/cm². For the optimization experiment (Supplementary Fig. 6A), two dyeing cycles were employed, the volume of the HCl solution was 0.45 mL/cm², and shaking was done on a laboratory platform shaker.

### CIELAB color space analysis

CIELAB color space (L*, a*, and b*) values were obtained using two measurement setups; a benchtop spectrophotometer (ColorFlex EZ,

HunterLab (USA)) equipped with a Xenon lamp (400–700 nm); and a custom spectral reflectance setup using an integrating sphere with 8°/ diffuse geometry (AvaSphere-50, Avantes (The Netherlands)) with a fibre coupled halogen light source (DH-2000, Ocean Insight (USA)) and a fibre coupled spectrometer (QE6500, Ocean Insight (USA)) controlled with AvaSofts software. In the custom setup the measurement area is a disc of diameter 5 mm and can be positioned over the whole swatch area. In both setups L*a*b* coordinates are calculated from measured spectral reflectance using a CIE 10° observer and D65 as the reference illuminant. Averages of three or five measurements in different positions are reported. CIELAB data from 60 swatches made from commercially purchased blue denim jeans (Supplementary Fig. 7) are shown as violin and boxplots with values from our experimental swatches overlaid, to easily compare the dying methods described in this study with data from real world jeans.

## TEA

Mass and energy balances were constructed for all processes based on the presented laboratory data and on literature[52–54] using MS Excel and process flow diagrams were constructed using commercial software called Lucidchart. Heat of reactions and thermodynamic equations were used for utility estimations for the chemical steps (A1–A6, Supplementary Fig. 2A). No utilities were considered for the enzymatic processes (B5-6). For the photolytic process, electricity was regarded as a raw material, not a utility. Due to the difference in technology readiness level (TRL) of the compared processes (commercial indigo (TRL 9); enzymatic indican (TRL 5); photolytic indican (TRL 2))[55] we used only raw materials and utilities to allow for a fair comparison of the production schemes. Process flow diagrams were constructed (Supplementary Fig. 2A), and industrial experts were consulted to validate necessary assumptions, including a yield of 90% for process step A5. Bulk prices for commodities and industrial averages for total production costs were used to construct cost tables for different scenarios (Supplementary Fig. 2B). The price of indoxyl (5 USD/kg) was estimated from the price of indigo, since indoxyl is not sold as a product per se and is present as an intermediate chemical for captive usage. The price of enzymes (25 USD/kg) was estimated conservatively (max. limit chosen) from previously published studies[56,57], and the import price of Lipase produced via *Aspergillus niger* fermentation. The prices of UDP (50 USD/kg) and UDP-Glc (84.9 kUSD/kg) were the cheapest available online (AliBaba and Carbosynth, respectively). The cost of water (0.0077 USD/kg) is based on proprietary information about RO purified water from BioBased Europe pilot facility in Belgium (http://www.bbeu.org/pilotplant). The prices of buffers and conventional dyeing reagents (0.1 USD/kg for citrate phosphate 90 mM, 246.5 USD/kg for HEPES powder, 0.059 USD/kg for phosphate 20 mM, 0.06 USD/kg for 30% HCl, 4.5 USD/kg for 27 w% caustic soda, 1.15 USD/kg for sodium hydrosulphite, 0.04 USD/kg for Setamol dispersing agent, 0.09 USD/kg for wetting agent) and sucrose (0.5 USD/kg) are those of bulk traded (4–200 kg, https://www.zauba. com). The price of electricity (0.12 USD/kWh) is that of european electricity (EU-28, www.statista.com). The indican raw material cost target was set to 60% of 3x the selling price of indigo (5 USD/kg). All assumptions and data sources for TEA are summarized in Supplementary Table 3. Further, open-source libraries in Python 3.9 were used to generate synthetic price distributions for Enzymes, Sucrose and Indoxyl (used in Supplementary Fig. 3G).

## LCA

The mass and energy balances generated during the TEA were input to comparative cradle-to-gate LCA, following the ISO14040 and ISO14044 standards for LCA[58]. The functional unit is 1 kg for dyestuff production and one pair of dyed jeans for dyeing at factory gate. Impacts of the mass and energy balances were assessed with the

SimaPro software with the Ecoinvent 3.6 database to provide information on the background processes (e.g. production of chemicals). The ReCiPe 2016 impact assessment methodology (hierarchist, v.1.02) was used both at midpoint and damage levels[59]. We compared 2 sets of processes: 1) dyestuff production (conventional indigo vs. chemoenzymatic indican), and 2) denim dyeing (conventional indigo dyeing, enzymatic indican dyeing, and photolytic indican dyeing) (Supplementary Table 2). Production impacts of all relevant chemicals were included. Impacts from potential chemical emissions (spills and leakages) were omitted, partly due to methodological gaps with respect to the toxicity of inorganics in available LCA inventories, and partly due to lack of publicly available, reliable, and quantitative data on well-known chemical spills from dyeing mills. This includes sodium hydrosulphite, which is one of the main reasons for this study. These necessary omissions likely lead to an underestimation of the environmental impacts of conventional indigo dyeing. LCA midpoints are reported in Supplementary Table 2 and in Figs. 4 and 5. The economic and environmental impact assessments were performed with consistent system boundaries. As demonstrated by Grasa and co-workers[60], TEAs are used to foresee the economic feasibility of conceptual and lab-based processes at an industrial scale. Based on the mass and energy balances of such TEA's, LCA's can be conducted to estimate environmental impacts at an equivalent scale[61–65]. The methodology was implemented here to convert to end points (Fig. 4A) and to a single sustainability score[21] (Supplementary Table 2). All assumptions and data sources for LCA are summarized in Supplementary Table 3.

## Figures

Molecular structures were made in Chemdraw. The non-colored icon of the jeans that are part of Fig. 1 is licensed by CC BY 4.0 and downloaded from https://www.onlinewebfonts.com/icon/472068.

## Statistics & reproducibility

No statistical method was used to predetermine sample size. No data were excluded from the analyses. The sample size of each experiment is given in the respective figure legend.

## Reporting summary

Further information on research design is available in the Nature Portfolio Reporting Summary linked to this article.

## Data availability

The authors declare that the data supporting the findings of this study are available within the paper and supplementary information files. Source data are provided with this paper. Other accession links: "Ecoinvent 3.6", PDB ID "5nlm", PDB ID "2vg8", PDB ID "2acv", PDB ID "6jtd", PDB ID "6lf6", PDB ID "7q3s" Source data are provided with this paper.

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

## Acknowledgements

The authors thank Carlotta Chiesa, Carsten Dam-Hansen, Dennis Dan Corell, Andreas Worberg, Nemeh Bani Odeh, and Gossa Garedew Wordofa for CIEL instrument time and technical assistance. This work was funded by the Novo Nordisk Foundation through grants NNF10CC1016517 (to the Novo Nordisk Foundation Center for Biosustainability), NNF20CC0035580 (to the Novo Nordisk Foundation Center for Biosustainability), NNF16OC0019088 (to D.H.W), and Carlsberg Foundation through grant CF18-0631 (to K.Q.).

## Author contributions

D.H.W. and O.Ö. conceived and designed the overall study. L.L.L. and G.B. designed variants of PtUGT1. D.T., K.Q. and C.U.J. designed the photochemistry study. G.B., D.T., F.F., C.U.J., E.P., and N.P. collected and analyzed the wet lab data. O.Ö., S.S. and A.-M.S. performed the sustainability assessments. D.H.W. drafted the manuscript. D.H.W., F.F., S.S., N.P., G.B. and D.T. designed and prepared the figures. All authors drafted relevant text parts, reviewed, and revised the manuscript and read and approved the final version.

## Competing interests

The authors declare the following competing interests: the Technical University of Denmark has submitted a patent application (application number WO2023161230A1) with inventors L.L.L., G.B., D.H.W. covering the *Pt*UGT1 enzyme variants. The Technical University of Denmark has submitted a patent application (application number EP22194865.6) with inventors N.P., G.B., D.H.W., K.Q., D.T., C.U.J., F.F. covering the photolytic dyeing process. The remaining authors declare no competing interests.
