## [Peer Review File · Nature Communications]

REVIEWER COMMENTS

Reviewer #2 (Remarks to the Author):

I appreciate the authors' revisions in response to reviewer comments. Responding on behalf of Reviewer #1 and #2, I have the following comments:

Regarding Reviewer #1's points 1-5 on sensitivity analysis, broader global impacts, social sustainability, the scalability of lab scale results, and data sources: I feel that the points have been addressed by the authors. It is especially helpful to see the data sources and assumptions that went into the LCA and TEA calculations in the Ecoinvent software.

Regarding Reviewer #1's point 6 on comparing photolytic vs enzymatic dyeing: My takeaway from Figure 6 is that enzymatic dyeing is better than photolytic dyeing on most metrics, but photolytic dyeing has the potential to be more economical than enzymatic dyeing, and both indican dyeing methods are better than conventional indigo. That seems to be the authors' message in the section starting from line 246. It could bolster the authors' point to show in Figure 6B the condition where one or two metrics are optimized (e.g. European energy grid and/or reducing energy usage to 0.24 kWh). That could show that with XYZ points of optimization, photolytic can be environmentally competitive to enzymatic (?), while being still cost-competitive to conventional (?).

I will skip Reviewer #1's point 7 because it looks like the figure in question has been removed/assimilated into existing figures.

Regarding Reviewer #1's point 8 about cost of resource categories: I actually don't think the corresponding figure legends are much different than before, but if the authors reference Extended Data Table 3 when talking about Fig. 4A and 5A (either in the main text or the figure legends), then in my opinion that would address the comment.

Regarding my previous comments, as Reviewer #2: I generally feel my comments were addressed. The only remaining comments I have mostly revolve around the previous Extended Data Figure 6, now Extended Data Figure 5:

Extended Data Figure 4C: Do you mean a particular pH here?

Extended Data Figure 5B: If all the samples were subject to light bulb treatment, why do you think you see the pH effect?

Extended Data Figure 5E: Do you know that all those other colors are indigo (or hydrolyzed indican)? Is it possible that the only one that turned into indigo is the one in HCl, possibly because it was treated with acid? I was mistaken about the reference on acid hydrolysis of indican being Song 2010. However, it is a previously known phenomenon (as Reviewer #3 also mentioned), and my group has actually also seen hydrolysis effects with acid on indican, independent of the presence of light.

Extended Data Figure 5G: That is helpful to see. Are both the light and dark conditions in the presence of HCl at pH 1.7, or are they in water?

Reviewer #3 (Remarks to the Author):

The authors addressed my concerns. I now support its publication in Nature Communications.

Point-by-point Response to Referees' comments on "Chemoenzymatic indican for light-driven denim dyeing" (NCOMMS-23-35211-T)

Dear editor and reviewers,
please find below a point-by-point response, which addresses all remaining reviewer comments. Also, we are happy to hear that Reviewer 2 finds that all Reviewer 1's points on sensitivity analysis, broader global impacts, social sustainability, the scalability of lab scale results, and data sources have been addressed in our resubmission.

Reviewer #2 (Remarks to the Author):

Regarding Reviewer #1's point 6 on comparing photolytic vs enzymatic dyeing: My takeaway from Figure 6 is that enzymatic dyeing is better than photolytic dyeing on most metrics, but photolytic dyeing has the potential to be more economical than enzymatic dyeing, and both indican dyeing methods are better than conventional indigo. That seems to be the authors' message in the section starting from line 246. It could bolster the authors' point to show in Figure 6B the condition where one or two metrics are optimized (e.g. European energy grid and/or reducing energy usage to 0.24 kWh). That could show that with XYZ points of optimization, photolytic can be environmentally competitive to enzymatic (?), while being still cost-competitive to conventional (?).

We appreciate the reviewer's interest and concur with their takeaway. Regarding the suggestion to add a scenario to the LCA with 100% green energy, we are a bit tentative, since we cannot argue that this is likely to happen for dyeing mills. Therefore, we would prefer to leave it with the current note, that 100% green energy would change the assessment significantly (line 257). Regarding the scenario with energy usage reduced to 0.24 kWh, we believe we already show its competitiveness in Extended Data Figure 6F. We have modified the description of these results to clarify that photolytic dyeing indeed would be cost-competitive with conventional dyeing and environmentally competitive with enzymatic dyeing if the energy usage could be reduced to 0.24 kWh/pair of jeans (line 254).

Regarding Reviewer #1's point 8 about cost of resource categories: I actually don't think the corresponding figure legends are much different than before, but if the authors reference Extended Data Table 3 when talking about Fig. 4A and 5A (either in the main text or the figure legends), then in my opinion that would address the comment.

We now reference Extended Data Table 3 in the figure legends of Fig. 4 and 6 (we believe the reviewer meant figure 6A instead of 5A, since 5A concerns the reaction mechanism and does not show any cost calculations).

Extended Data Figure 4C: Do you mean a particular pH here?

We thank the Reviewer for pointing out the missing information, this was a mistake which has now been corrected. The optimal pH is 9.

Extended Data Figure 5B: If all the samples were subject to light bulb treatment, why do you think you see the pH effect?

Indeed, all samples were subjected to the same light bulb exposure. In conditions below pH 1.7, we probably see acid hydrolysis with by-product formation. Also, alkaline conditions (pH 9 and 11) most likely result in unspecific cleavage (probably hydrolysis) and therefore side reactions. The same side reactions (leading to formation of a light purple coloured side product) were observed in the solvent study using basic condition (Extended Data Figure 5C). We have added a sentence to clarify this (line 219).

Extended Data Figure 5E: Do you know that all those other colors are indigo (or hydrolyzed indican)? Is it possible that the only one that turned into indigo is the one in HCl, possibly because it was treated with acid? I was mistaken about the reference on acid hydrolysis of indican being Song 2010. However, it is a previously known phenomenon (as Reviewer #3 also mentioned), and my group has actually also seen hydrolysis effects with acid on indican, independent of the presence of light.

There are no colors in Extended Data Figure 5E (see next paragraph). Like the reviewer and other researchers, we did observe acid hydrolysis of indican at high HCl (aq.) concentrations (1M and higher), with simultaneously biproduct formation. However, we did not see any indican degradation in aqueous 20 mM HCl (pH1.7) conditions in the dark (Extended Data Figure 5G). These results demonstrate that both 20 mM HCl and light are necessary for clean indigo formation. We did not further analyse the minor impurities observed in the data presented in Extended Data Figure 5E, but instead continued with the reaction resulting in clean formation of indigo.

Perhaps the reviewer is here also referring to Extended Data Fig. 5C with the colors. Indeed, it looks like the only one that efficiently turned into indigo is the one in HCl, since indigo is blue. It is well known that indoxyl can form other compounds with other colors, e.g. indirubin. However, this does not change the fact that we see no indigo formation in 20 mM HCl in the absence of light treatment (Extended Data Figure 5G).

Extended Data Figure 5G: That is helpful to see. Are both the light and dark conditions in the presence of HCl at pH 1.7, or are they in water?

The reactions in Extended Data Figure 5G were done in aqueous 20 mM HCl conditions (pH 1.7) in the dark and light, respectively. The conditions are now described in the Figure legend, in addition to in the Methods section as before.

Reviewer #3 (Remarks to the Author):

The authors addressed my concerns. I now support its publication in Nature Communications.

We thank the reviewers for their time.

REVIEWERS' COMMENTS

Reviewer #2 (Remarks to the Author):

The authors have addressed my concerns, and I support publication.